# The Conserved Non-Coding Sequence 2 (CNS2) Enhances CD69 Transcription through Cooperation between the Transcription Factors Oct1 and RUNX1

**DOI:** 10.3390/genes10090651

**Published:** 2019-08-28

**Authors:** Miguel G. Fontela, Laura Notario, Elisenda Alari-Pahissa, Elena Lorente, Pilar Lauzurica

**Affiliations:** 1Microbiology National Center, Instituto de Salud Carlos III, Majadahonda, 28220 Madrid, Spain; 2Department of Experimental and Health Science, University Pompeu Fabra, 08003 Barcelona, Spain

**Keywords:** CD69, transcriptional regulation, transcription factor, enhancer, noncoding regions

## Abstract

The immune regulatory receptor CD69 is expressed upon activation in all types of leukocytes and is strongly regulated at the transcriptional level. We previously described that, in addition to the *CD69* promoter, there are four conserved noncoding regions (*CNS1-4*) upstream of the *CD69* promoter. Furthermore, we proposed that *CNS2* is the main enhancer of *CD69* transcription. In the present study, we mapped the transcription factor (TF) binding sites (TFBS) from ChIP-seq databases within *CNS2*. Through luciferase reporter assays, we defined a ~60 bp sequence that acts as the minimum enhancer core of mouse *CNS2*, which includes the Oct1 TFBS. This enhancer core establishes cooperative interactions with the 3′ and 5′ flanking regions, which contain RUNX1 BS. In agreement with the luciferase reporter data, the inhibition of RUNX1 and Oct1 TF expression by siRNA suggests that they synergistically enhance endogenous *CD69* gene transcription. In summary, we describe an enhancer core containing RUNX1 and Oct1 BS that is important for the activity of the most potent *CD69* gene transcription enhancer.

## 1. Introduction

In mammals, 95% of the genome is noncoding, and 40% of the noncoding region plays a role in transcription regulation [1]. In combination with gene promoters, cis-regulatory elements contribute to gene transcriptional regulation as enhancers or repressors. Cis-regulatory elements are often identified as conserved noncoding sequences (*CNS*) with DNase I hypersensitivity. Cis-regulatory elements contain binding sites for trans-acting transcription factors (TFBSs) and serve as centers of epigenetic changes [2,3]. These elements can be located around or within genes and can affect their transcription from as far as megabases [4]. The three-dimensional conformation of the genome seems to be important to promote contacts between distant elements and their target genes. Thus, these sequences form complex regulatory landscapes within the noncoding genome.

CD69 is a leukocyte activation marker that plays an important role in the regulation of immune responses [5,6,7,8]. CD69 is dully expressed on the membrane of some leukocyte subsets in a steady state [9,10,11], and it is strongly and rapidly upregulated on all leukocytes upon activation. By the study of CD69-deficient mice, CD69 has been defined as a regulator of the immune response in different murine models of tumor, infection, autoimmune disease and other inflammatory models [6,7,8,12,13,14,15,16]. Galectin-1, S100A8/A9 and myosin light chains 9 and 12 have been proposed as CD69 ligands [5,17,18]. In T cells, CD69 is upregulated in response to TCR engagement [19,20], cytokines [21,22,23] and PKC activators [11,24,25,26,27]. This upregulation is strongly dependent on transcriptional induction [20,25,28]. Human and mouse *CD69* promoters are able to direct transcription in resting cells, as well as to increase *CD69* transcription in stimulated cells [29]. This inducibility is attributed to cis-elements located in the promoter interacting with Erg-1, Erg-3, ATF-3/CREB, AP-1 and NFkB transcription factors [29,30,31].

We previously described the existence of, in addition to the *CD69* promoter, another five cis-regulatory regions: A non-conserved hypersensitivity site (HS) located within the first intron of the *CD69* gene and four conserved noncoding regions (*CNS1-4*) upstream of the *CD69* promoter [32,33]. In murine T cells, these regions showed DNaseI sensitivity and specific histone modifications. Moreover, *CNS2* and *CNS4* had constitutive and inducible enhancer activity in reporter assays, with *CNS2* being the region with the highest inducible enhancer capacity [32,33]. Furthermore, in a recent work, we proposed that *CNS2* is the main enhancer of *CD69* promoter transcription in vivo [34] based on the characterization of TFBSs conserved between 6 mammal species and explored the contribution of some of its regulatory features to its enhancing capacity. However, that study did not take into account sequences to which transcription factors bind. In the present study, we mapped TFBSs on mouse *CNS2* according to updated ChIP-seq data; and, based on their distribution, we divided them into five different regions. Then, we determined the enhancer activity of *CNS2* and identified its location through luciferase reporter assays. With this approach, we found a minimum enhancer core of ~60 bp that includes an Oct1 (*POU2F1*) BS. This core functionally interacts with adjacent regions at the 5′ end, which contains RUNX1 BSs, and at the 3′ end, which contains SRF BSs, via a cooperative modular model that leads to *CD69* promoter induced enhancement of transcription. Moreover, we assessed the roles of Oct1 and RUNX1 in the transcriptional regulation of the endogenous *CD69* gene by siRNA. In agreement with the luciferase reporter data, the gene silencing results also indicate synergistic cooperation between these TFs to enhance *CD69* transcription.

In summary, our work strengths support indicating that *CNS2* is a potent enhancer of mouse *CD69* promoter transcription and defines an enhancer core that acts together with flanking cis-interacting TFs through a cooperative modular model to increase *CD69* promoter induced enhancement of transcription.

## 2. Materials and Methods

**Data-mining, identification and mapping of conserved TFBSs within *CNS2*.** The *CNS2* sequences of several mammal species were downloaded and analyzed for the presence of conserved TFBSs, as previously described [34]. Transcription factor binding data determined by chromatin immunoprecipitation followed by sequencing (ChIP-seq) in different cell lines were obtained from the ENCODE consortium for human (hg19 genome) data [35] and both the ChIP Atlas [36] (Database Center for Life Science, Kyushu University, Fukuoka, Japan) and the ChIPBase v2.0 (version 2.3.4, Qu Lab, School of Science, Sun Yat-Sen University, Guangzhou, China) for mouse (mm10 genome) data [37] and were displayed using the SnapGene^®^ software (GSL Biotech; available at www.snapgene.com) to create a comprehensive map of the TFBSs contained within *CNS2*.

**Mouse luciferase plasmids.** The mouse *CD69* promoter (−1 to −609, BAC clone RP24-188C4) was cloned into BglII and HindIII restriction enzyme (RE) cloning sites of the commercial luciferase vector pGL3 basic (Promega, E1751, Madison, WI, USA, [Addgene sequence]) (pPr plasmid). Then, the *CNS2* region (mouse 2010 chr6: 129,234,359–129,235,318) was cloned into KpnI and XhoI RE sites, with the introduction of an EcoRI site by KpnI for further cloning (pPr2 plasmid). Modified *CNS2* constructs containing different regions or specific TFBSs were generated by PCR amplification of pPr2 employing custom primers (Table 1), followed by cloning of the fragments into EcoRI and XhoI sites in the plasmid containing the mouse *CD69* promoter (Table 2). The resulting luciferase plasmids were validated through Sanger sequencing (Figure 1).

**Luciferase assays.** Human Jurkat T cells (5–7 × 10^5^) were transfected with 1 µg of modified firefly luciferase plasmid (purified with the Plasmid Maxi Kit from Qiagen 12163) plus 20 ng of pRL-TK (Renilla luciferase plasmid, Promega E224, Madison, USA [Addgene sequence]) to standardize the luciferase activity independent of the efficiency of transfection between samples) using Effectene (Qiagen 301425) following the manufacturer’s protocol. After transfection, the cells were cultured at 37°C with 5% CO_2_ for 24 h and were then stimulated with 10 ng/mL PMA and 500 ng/mL Ionomycin for an additional 24 h. Forty-eight hours after transfection, cells were lysed with passive lysis buffer (Promega E1941, Madison, WI USA), and luciferase activity (firefly/renilla) was measured with a dual luciferase kit (Promega E1910, Madison, WI USA) using the Orion II microplate luminometer (Berthold 11300010, Bad Wildbad, Germany).

**Design of dicer substrate small interfering RNA (DsiRNA).** All DsiRNAs described were synthetized by IDT (Integrated DNA Technologies, Coralville, IA, USA). Three DsiRNAs were designed for each gene to be silenced in addition to two additional DsiRNAs and a negative control targeting NC1. The genome reference used for DsiRNA design was mm10. Table 3 provides the sequence and cross-reacting properties of DsiRNAs against POU2F1 (Oct1) and RUNX1 mRNAs.

**DsiRNA transfection and CD69 characterization.** Mouse EL-4 T cells (10^5^) were transfected with a 1 µM pool of three DsiRNAs against *POU2F1* (Oct1) or *RUNX1* in a 1:1:1 ratio or a negative control DsiRNA against NC1. RNA of transfected cells were extracted, and a qPCR was performed to assess mRNA levels of *RUNX1* and *POU2F1* (Oct1).

Transfected cells were cultured in a 96-well plate at 37 °C with 5% CO_2_ for 16 h and then stimulated or not with 10 ng/mL PMA and 500 ng/mL ionomycin for an additional 6 h. Twenty-two hours after transfection, CD69 surface expression was measured by flow cytometry as follows: cells were washed twice in cold 1X PBS and stained for 30 min at 4 °C with 0.5 µg of an α-mCD69/PE-Cy7 monoclonal antibody (Clone H1.2F3, eBioscience 25-0691-81, San Diego, CA, USA). Samples were acquired with a FACSCanto (Becton Dickinson, Franklin Lakes, NJ USA) flow cytometer, and data were analyzed using FlowJo software. The percentage of inhibition of CD69 expression with respect to its expression in the NC-1 control was represented using GraphPad Prism 7 (version 7, San Diego, CA, USA).

**Data representation and statistical analysis.** Luciferase assay and flow cytometry data were plotted with GraphPad Prism 7. Significant differences between multiple conditions were tested with a nonparametric one-way ANOVA test performed with GraphPad Prism 7.

## 3. Results

### 3.1. Mapping of Transcription Factor Binding Sites within Mouse CNS2 Enhancer

In a previous data-mining study of the regulatory features of the *CNSs* of the *CD69* gene, we proposed that *CNS2* is the putative main enhancer based on the enrichment of chromatin accessibility, modifications and bound TF [34]. Given the availability of new Chip-seq data, in the present work, we updated the map of TFBSs in the human locus using Chip-seq data from human hematopoietic cell lines available from the ENCODE consortium (Figure 2A) and also mapped bound TF to the mouse locus using data from murine primary B and T cells available from the ChIP Atlas and ChIPBase v2.0 databases (Figure 2B) to compare them. These maps showed that the human *CD69* 3′untranslated region, intron 1, promoter, *CNS1*, *CNS2* and *CNS4* and in the murine 3′untranslated region, promoter, *CNS1* and *CNS2* have a higher density of TFBSs (TFBSs per 100 bp) than the rest of noncoding sequence within the *CD69* locus. Thus, the 3′untranslated region, promoter, *CNS1* and *CNS2* of both species contain numerous TFBSs, which suggests that they may play a particularly important role in the transcriptional regulation of the *CD69* promoter. In agreement with our previous observations, mouse *CNS2* also shows a higher total number of TFBSs compared to the other regulatory regions within *CD69* locus.

In our previous work [34], we subdivided *CNS2* into different subregions based on the presence of TFBSs conserved between 6 mammal species and tested their contribution to the enhancer capacity of *CNS2*. However, that characterization left undefined regions where bound TFs have subsequently been identified and could play a relevant role. Moreover, considering that some regulatory mechanisms may have diverted during evolution and that, consequently, some important TFBS might not be conserved, in the present work, we subdivided the mouse *CNS2* into five regions based on the abovementioned updated ChIP-seq data, taking into account all bound TFs, as well as those bound to sites that are not conserved between six mammal species (Figure 3A). Regions II and III contain more than 70% of TFs bound to non-conserved sites, while regions I and IV contains TFs bound only to conserved BSs. Region V has intermediate characteristics.

### 3.2. Transcriptional Enhancer Capacity of the CNS2 Region and Its Enhancer Core Activity

Then, we assessed the contribution of the different regions to mouse *CNS2* enhancer capacity using a luciferase reporter assay in Jurkat cells stimulated or not with PMA plus ionomycin. In this experiment, we compared the capacity of the complete *CNS2* sequence (Prom+I-V) to enhance transcription from the mouse *CD69* promoter with those of different *CNS2* fragments: I+II+III, II+III and I+II (Prom+I-III, Prom+II-III and Prom+I-II). As shown in Figure 3B regions II+III have an inducible enhancer activity similar to that of regions I+II+III, while regions I+II have lower enhancer potential. Therefore, regions II+III seem to contain the most of the relevant enhancing features for transcription for the *CD69* promoter upon PMA/ionomycin activation, while those of regions I, IV and V have a more modest contribution.

While region III contains three overlapping BSs, region II has two separate subregions with TFBSs: One with an Oct1 BS and another with GATA+MyoD TFBSs. We further assessed the contribution of the individual regions I, II and III, as well as the subregions of region II separately or in combination with the region I or III, but keeping, within the tested fragment, the original sequence with the endogenous TFBSs distribution. Figure 4 shows that region II (Prom+II) significantly increases CD69 promoter transcriptional activity upon activation, but neither region I (Prom+I) nor region III (Prom+III) increases CD69 promoter transcriptional activity upon activation (Figure 4B).

The ~60 bp sequence of region II containing the Oct1 TFBS (Prom+Oct1) significantly increases the inducible enhancer activity in a similar manner as region II (Prom+II), while the ~65 bp sequence containing GATA+MyoD TFBSs (Prom+GATA+MyoD) did not significantly increase the inducible enhancer activity. These results suggest that the subregion that contains the Oct1 TFBS constitutes the most relevant regulatory feature of region II. Moreover, considering that regions I and III alone do not show enhancing activity, these data indicate that the ~60 bp sequence containing the Oct1 TFBS is the shortest sequence identified within regions I+II+III that has transcriptional enhancer activity, likely acting as the enhancer core of *CD69* transcription. Furthermore, the combination of this sequence with region I (Prom+I+Oct1, ~200 bp) had a transcriptional enhancing capacity as high as half of that of region I-III and similar to that of regions I and II together (Figure 3B and Figure 4B). Although the combination of region III with the ~65 bp sequence GATA+MyoD (Prom+GATA+MyoD+III, ~190 bp) increased the transcriptional activity twofold compared to the sequences containing the BS of each TF individually, it did not reach transcription levels significantly higher than those of the promoter alone. Since GATA and MyoD BS are between Oct1 and RUNX1, it is not possible to test Oct1+RUNX1 without altering the original TFBSs distribution. Thus, considering that regions II+III have almost as much enhancer activity as regions I-III (Figure 3B), it is suggested that the Oct1 TFBS also needs to cooperate with region III to enhance transcription. Thus, the Oct1 TFBS seems to act as the enhancer core of *CD69* transcription and can synergize with either region I or region III or with both of them to achieve greater enhancement of transcription from the CD69 promoter. Altogether, these data suggest a cooperative modular model between the TFs in region II and those in regions I and III.

### 3.3. RUNX1 and Oct1 Silencing Affects the Transcriptional Regulation of CD69

Given the synergy observed between the Oct1 BS and region III and the fact that the latter contains a BS for RUNX1, an important TF during development and homeostasis of immune cells, we evaluated the relative contribution of Oct1 and RUNX1 to *CD69* transcriptional regulation. To do so, we silenced them individually or in combination using siRNA in the mouse T cell line EL-4 and analyzed the effect of silencing on surface CD69 protein in unstimulated cells or cells activated with PMA/Ion. Under both conditions, inhibition of RUNX1 and Oct1 individually resulted in a CD69 expression reduction of ~40%, and the joint inhibition of both TFs did not further reduce CD69 expression (Figure 5). Therefore, RUNX1 and Oct1 contribute to enhancing *CD69* expression both at resting and after stimulation. Moreover, the fact that the effect of their individual silencing was the same as that of their combined deletion suggests that they need each other to increase *CD69* expression, which supports the cooperative model of enhancing transcription.

## 4. Discussion

In this work, we deepened the understanding of the enhancer function of mouse conserved noncoding sequence 2 (*CNS2*) on *CD69* promoter transcriptional regulation. Using ChIP-seq data, we mapped the TFBS within *CNS2* and used the TFBS distribution to define different regions. Then, we tested the effect of these regions on transcription induced by the *CD69* promoter in response to PMA/ionomycin activation in luciferase reporter assays. In this way, we defined a minimal sequence with intrinsic enhancer capacity that contained a conserved Oct1 BS that could further enhance transcription if it was accompanied by either a 3′ flanking region containing a conserved SRF BS or a 5′ flanking region containing conserved RUNX1, GABPA and Elk-1 TFBSs and could enhance transcription to even higher levels in the presence of both of them. These results suggest cooperation between these TFs. We further tested the effect of Oct1 and RUNX1 silencing on CD69 protein expression. It these experiments, the observed synergy seemed even more striking because the effect of each TF depended on the presence of the other.

In previous studies, we observed that among the different *CNSs*, *CNS2* had the most potent enhancer activity on *CD69* transcription. In the present work, the mapping of TFBSs on the *CD69* locus showed that several of the TFs that have been shown to bind to *CNS2* also bind the promoter region, which suggests their importance in the transcriptional regulation of *CD69*. Despite the fact that there are fewer data available for the TFs bound to the mouse *CD69* locus than to the human *CD69* locus, similar to humans, mouse *CNS2* is also the region with the highest enrichment of TFBSs. Occupancy by specific TF has been described to be a predictor of enhancer activity of a given region [38]. Thus, these results highlight the importance of *CNS2* as a transcriptional enhancer of *CD69*.

The present results regarding the enhancer capacity of *CNS2* regions are in agreement with our previous work, in which the regions with the highest enhancer capacity comprises regions I, II and III, described in the current study [34]. In the present study, the further subdivision of this regions into smaller regions allowed us to identify an Oct1 BS-containing minimum enhancer core that can cooperate with either an adjacent region at the 3′ end (region I, containing an SRF TFBS) or 5′ end (region III, containing a RUNX1 TFBS) or with both of them to reach greater enhancer activity. Interestingly, although region I did not have enhancer activity *per se*, it could increase that of Oct1 BS. This could be explained by an interaction of TFBSs in region I with the transcriptional machinery that bounds to the region containing the Oct1 BS. Altogether, these data suggest a cooperative modular model of transcription factors that synergistically enhances transcription. Cooperation between transcription factors may occur based on their physical association, a mechanism frequently described for enhancer sequences that affect the transcriptional behavior of several immune system genes [39,40,41].

Although Oct1 has been widely studied as a regulator of transcription in many cell types [42,43], its precise role is still unclear. Oct1 is known to have both activating [44,45,46] and silencing activity [47,48,49]; moreover, this TF has been described to play a role in CD4^+^ T cell differentiation [50] and memory formation [51], orchestrating the fate and function of CD4^+^ effector T cells as a switchable stabilizer of the repressed and inducible state [52], acting at hypersensitivity sites that are distant from promoters of target genes in several cases. RUNX1 has been implicated in the development and homeostasis of different immune subpopulations [53].

As is the case for *CNS2*, many enhancers are located far from their target genes, but are able to specifically communicate with distal promoters over large distances [54,55,56] through the formation of a chromatin loop [57,58]. Thus, chromosome topology is critical for the interactions between distant enhancers and promoters. However, the relevance of particular interactions for target gene transcription is difficult to assess because they have not always been related to functional outcomes [59,60].

Because of the lack of a genomic context in reporter plasmids, luciferase assay data do not reflect the effects of chromosome topology on transcription. However, the luciferase assay can still be used as a predictor of the regulatory capacity of elements that can be further characterized in a more physiologic context. With that aim, we tested the effects of RUNX1 and Oct1 silencing on the expression of the endogenous *CD69* gene. Silencing of these TFs separately in the mouse EL-4 T cell line reduced CD69 expression as much as their joint silencing, suggesting that they can enhance CD69 expression, but only if the other TF is also present. This result suggests a mode of action that is completely synergic. Given the predominant enhancer role of *CNS2* in *CD69* transcription and, within it, of the Oct1 TFBS, and its observed cooperation with region I, we believe that these results might reflect, to some extent, the role of Oct1 and RUNX1 acting on the BSs described in regions I and II of *CNS2*. However, we cannot rule out that they might also be influenced by the action of these TFs on other BSs within the *CD69* locus. The ChIP-seq databases show two RUNX1 BS found in *CD69* promoter sequence. However, since these regions have a much more modest enhancing capacity, we propose that the main enhancer effect of RUNX1 is through the BS in *CNS2*. Future experiments using ChIP, TF silencing and TFBS mutagenesis might shed more light on the role of the studied TF in *CD69* transcription in the genomic context.

In summary, our results strengthen evidence of the role of *CNS2* as an essential enhancer of mouse *CD69* promoter transcription. Despite the need for further studies on the role of *CNS2* in the genomic context, to our knowledge, no detailed studies on the composition and function of regulatory elements within this enhancer have been published. Our work is, therefore, a step forward in understanding how *CNS2* regulates *CD69* expression.

## Figures and Tables

**Figure 1 genes-10-00651-f001:**
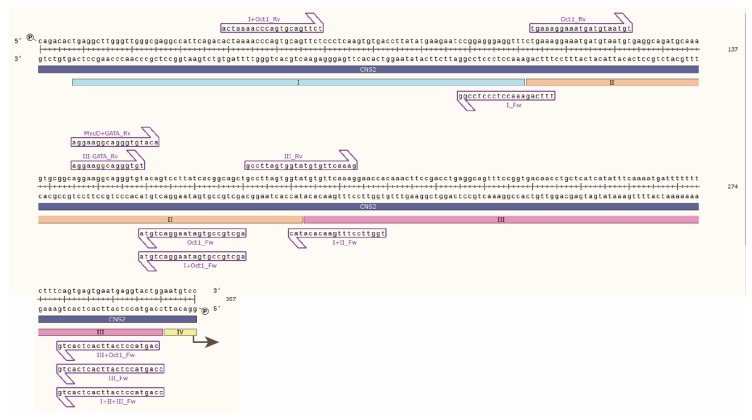
Sanger sequencing of mouse *CNS2* sequence containing regions I, II and III. Primers used to amplify the *CNS2* sequences used in the luciferase experiments are depicted as purple arrows.

**Figure 2 genes-10-00651-f002:**
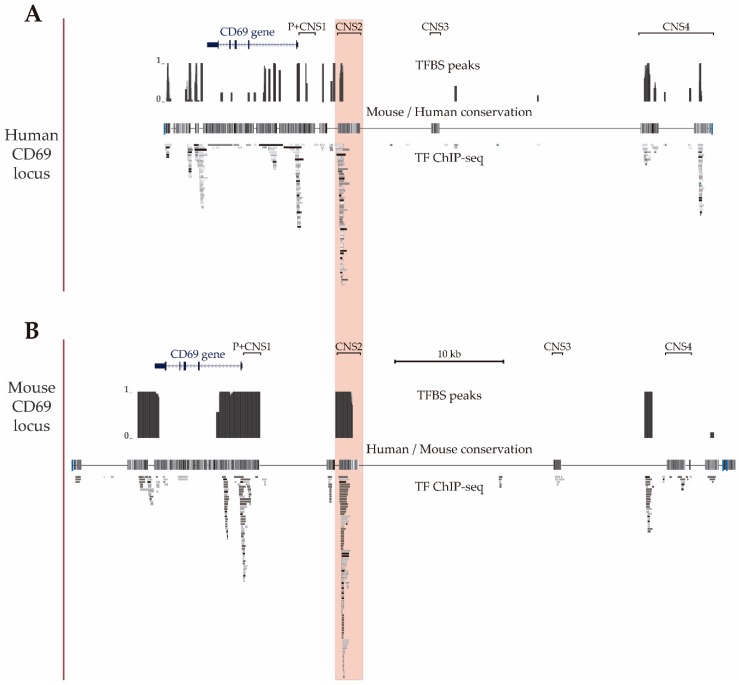
CD69 gene locus. UCSC genome browser showing the human (**A**) (GRCh37/hg19 assembly, chr12: 9,900,752-9,952,771) and mouse (**B**) (GRCm38/mm10 assembly, chr6: 129,259,289–129,321,308) CD69 locus and the four upstream conserved noncoding regions (CNS1-4). The alignment of the mouse and human genomes are displayed as a grayscale density plot that indicates the alignment quality, where darker values indicate higher levels of overall conservation. Transcription factor binding site peaks (TFBS peaks) show the proportion (between 0 and 1) of ChIP-seq data sets reporting TF binding at each location. The grey boxes represent different transcription factors found by ChIP-seq in human cell lines H1-hESC, A549, GM12878, HeLa-S3, IMR90, K562, HepG2, MCF-7, SK-N-SH, SK-N-SH_RA, HUVEC, AG04449, AG04450, AG09309, AG09319, AG10803, AoAF, BE2_C, BJ, Caco-2, Dnd41, ECC-1, Fibrobl, GM06990, GM08714, GM10847, GM12801, GM12864, GM12865, GM12872, M12873, GM12874, GM12875, GM12891, GM12892, GM15510, GM18505, GM18526, GM18951, GM19099, GM19193, GM19238, GM19239, GM19240, Gliobla, HA-sp, HAc, HBMEC, HCFaa, HCM, HCPEpiC, HCT-116, HEEpiC, HEK293, HEK293-T-REx, HFF, HFF-Myc, HL-60, HMEC, HMF, HPAF, HPF, HRE, HRPEpiC, HSMM, HSMMtube, HVMF, MCF10A-Er-Src, NB4, NH-A, NHDF-Ad, NHDF-neo, NHEK, NHLF, NT2-D1, Osteobl, PANC-1, PBDE, PBDEFetal, PFSK-1, ProgFib, RPTEC, Raji, SAEC, SH-SY5Y, SK-N-MC and T-47D (available at UCSC) and in mouse primary T CD4^+^, T CD8^+^ and B lymphocytes (TF-ChIP-seq, available at ChIP Atlas and ChIP-Base v2.0).

**Figure 3 genes-10-00651-f003:**
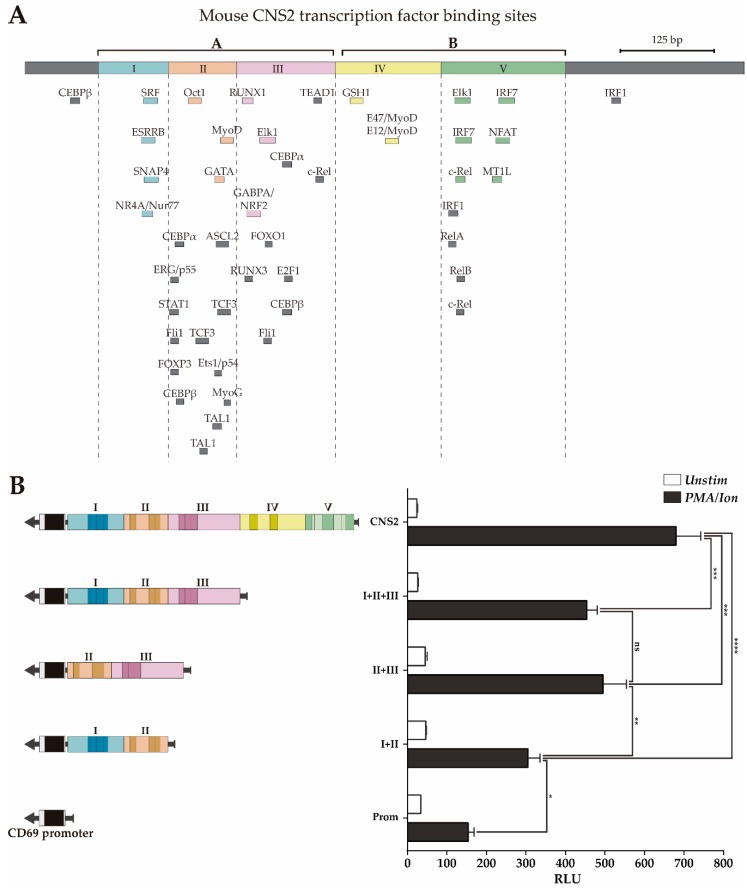
Luciferase activity of *CNS2* regions II+III is similar to that of the whole *CNS2* sequence. (**A**) Map of the *CNS2* regions and TFBS defined by ChIPSeq data. Color boxes indicate transcription factor binding sites conserved among six mammals species, as previously described [34]; grey boxes indicate non-conserved TFBSs identified by public ChIP-seq experiment databases. (**B**) Jurkat cells were transfected with luciferase constructs carrying the mouse *CD69* promoter alone or together with the whole *CNS2* or regions I+II, II+III or I-III. The bars show the mean RLU of three independent experiments. The error bars show the standard deviation. One-way ANOVA was applied; * (*p* < 0.05), ** (*p* < 0.01), and *** (*p* < 0.001). **RLU:** Relative luciferase units.

**Figure 4 genes-10-00651-f004:**
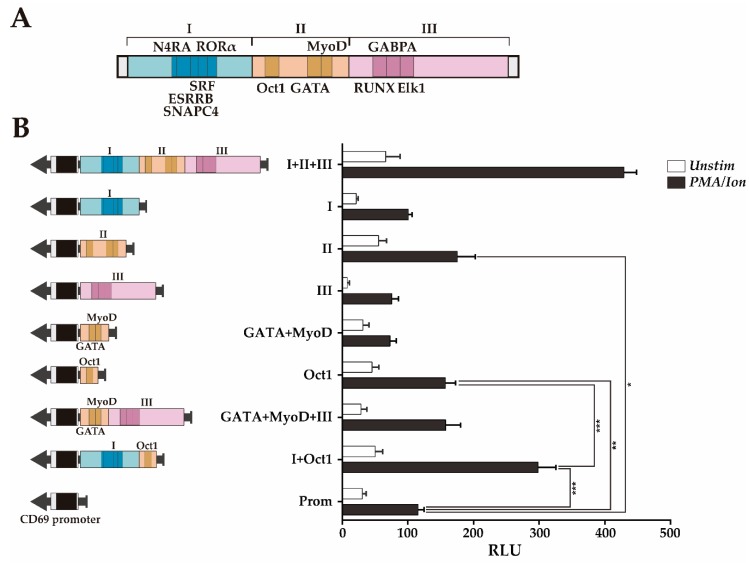
Transcriptional enhancer activity of the *CNS2* regions I+II+III and region II subregions. (**A**) Map of regions I-III and the conserved TFBSs within them. (**B**) Measurement of the luciferase activity after stimulation of Jurkat cells transfected with constructs carrying the mouse *CD69* promoter alone or in combination with the different regions and subregions (I+II+III, I, II, II, GATA+MyoD, OctI, GATA+MyoD+III, Oct1+I). The bars show the mean RLU. The error bars show the standard deviation of three independent experiments with duplicate measurements. One factor ANOVA was applied to test for significant differences, with * (*p* < 0.05), ** (*p* < 0.01), and *** (*p* < 0.001). **RLU:** Relative luciferase units.

**Figure 5 genes-10-00651-f005:**
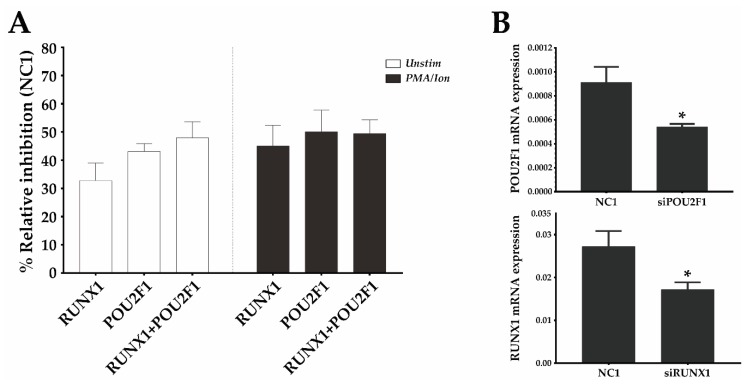
RUNX1 and Oct1 (*POU2F1*) silencing downregulates mouse CD69 protein expression. (**A**) EL-4 T cells were transfected with a 1 µM pool of three DsiRNAs against *RUNX1*, POU2F1 (Oct1), a combination of both, or a non-targeting DsiRNA (NC-1) and were cultured for 16 h. They were then stimulated or not stimulated with PMA/Ion, and the CD69 surface levels were measured 6 h later. The bars represent the mean ± SEM of the % relative inhibition with respect to CD69 expression in the NC-1 control of 2 different experiments in which each transfection was performed in duplicate. (**B**) mRNA expression of *RUNX1* and *POU2F1* was analyzed by qPCR 22 h after DsiRNA transfection, compared with the NC1 DsiRNA control. Data are presented as means ± SD of two independent experiments (* *p* < 0.05).

**Table 1 genes-10-00651-t001:** List of the primers used to amplify the sequences within *CNS2* for further cloning and sequencing. The sequence in italics indicates the extra bases that allow restriction enzymes to cut; bold sequences indicate XhoI and EcoRI target sequences; plain text indicates the *CNS2* sequence.

Primer	Sequence 5′ → 3′
**I_Fw**	*GAG***GAATTC**TTTCAGAAACCTCCCTCCGG
**I_Rv**	*AGA***CTCGAG**CAGCATCTGTTGTGATTAGCAT
**I+II+III_Fw**	*CC***GAATTC**CCAGTACCTCATTCACTCACTG
**I+OCT_Fw**	*AAAA***GAATTC**AGCTGCCGTGATAAGGACTGTA
**III_Fw**	*ATAAA***GAATTC**CCAGTACCTCATTCACTCACTG
**I+OCT_Rv**	*AGT***CTCGAG**CACTAAAACCCAGTGCAGTTCT
**III+OCT_Fw**	*AAA***GAATTC**CCAGTACCTCATTCACTCACTG
**OCT_Fw**	*AAAA***GAATTC**AGCTGCCGTGATAAGGACTGTA
**OCT_Rv**	*AAA***CTCGAG**CTGAAAGGAAATGATGTAATGT
**III_Rv**	*AAA***CTCGAG**GCCTTAGTGGTATGTGTTCAAAG
**I+II_Fw**	*AAAA***GAATTC**TGGTTCCTTTGAACACATAC
**III-GATA_Rv**	*AAAA***CTCGAG**AGGAAGGCAGGGTGT
**MyoD+GATA_Rv**	*AAA***CTCGAG**AGGAAGGCAGGGTGTACA
**RV3_Fw ***	CTAGCAAAATAGGCTGTCCC
**Prom_Seq_Rv ***	TGACATGGGAAAAGCACTGGA

* indicates the primers used for sequencing and validation of the resulting plasmids.

**Table 2 genes-10-00651-t002:** Primers used for the amplification of luciferase plasmids, and length of the expected amplicon. Below is displayed the primers in the DNA sequence of regions I–III.

Construct	Primers Used	Amplicon Length
**I+II+III**	I+II+III_Fw + I + Oct1_Rv	297 bp
**II+III**	III+Oct1_Fw + Oct1_Rv	227 bp
**I+II**	Oct1_Fw + I + Oct1_Rv	177 bp
**I**	I_Fw + I+Oct1_Rv	70 bp
**II**	I+II_Fw + Oct1_Rv	107 bp
**III**	III_Fw + III_Rv	120 bp
**GATA+MyoD**	I+II_Fw + MyoD + GATA_Rv	65 bp
**Oct1**	Oct1_Fw + Oct1_Rv	60 bp
**GATA+MyoD+III**	III_Fw + III-GATA_Rv	156 bp
**I+Oct1**	I+Oct1_Fw + I+Oct1_Rv	142 bp

**Table 3 genes-10-00651-t003:** Nucleotide sequences of the sense and antisense strands and cross-reaction of the DsiRNAs used to silence RUNX and *POU2F1* (Oct1).

*DsiRNA*	Sequences	Cross-Reacting Transcript
**Runx1.13.1**	**5′**-AAGAAAGAUAUCAAGUACUACAUtt-**3′** **3′**-AAUUCUUUCUAUAGUUCAUGAUGUAAA-**5′**	NM_001111022 NM_001111021 NM_001111023 NM_009821
**Runx1.13.2**	**5′**-AUGGCAGGCAACGAUGAAAACUAct-**3′** **3′**-AGUACCGUCCGUUGCUACUUUUGAUGA-**5′**	NM_009821 NM_001111023 NM_001111021 NM_001111022
**Runx1.13.3**	**5′**-GAAGAACCAGGUAGCGAGAUUCAac-**3′** **3′**-UACUUCUUGGUCCAUCGCUCUAAGUUG-**5′**	NM_001111022 NM_001111021 NM_001111023 NM_009821
**Pou2f1.13.1**	**5′**-UAAAUUUCAUGAAAGCUUUACUUgt-**3′** **3′**-CGAUUUAAAGUACUUUCGAAAUGAACA-**5′**	NM_198932 NM_011137 NM_198934 NM_198933
**Pou2f1.13.2**	**5′**-CAAGAAUGAAUAAUCCAUCAGAAac-**3′** **3′**-GAGUUCUUACUUAUUAGGUAGUCUUUG-**5′**	NM_198933 NM_198934 NM_011137 NM_198932
**Pou2f1.13.3**	**5′**-GCAGUUUGCCAAGACUUUCAAACaa-**3′** **3′**-CUCGUCAAACGGUUCUGAAAGUUUGUU-**5′**	NM_198932 NM_011137 NM_198934 NM_198933

Upper case letter represent *RNA* bases, while lower case letters correspond to *DNA* bases.

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
