# Peer review of "The Conserved Non-Coding Sequence 2 (CNS2) Enhances CD69 Transcription through Cooperation between the Transcription Factors Oct1 and RUNX1"

_genes, 2019, doi:10.3390/genes10090651_

Round 1

Reviewer 1 Report

This manuscript looks like a follow-up of authors' another study published in 2015 (ref: 27). In the paper already published, they described the existence of five ci-regulatory regions on conserved non-coding sequences (CNS) 1-4 to promote CD69. Now the authors tried to link two factors, Oct1 and RUNX1, on the CNS2 to CD69. The idea sounds interesting but the evidence  presented in this version seems quite weak. Below are my major concerns about the results:

(1) Luciferase activity experiments were done to identify the functional region(s) on CNS2. Figure 2 shows that region I might not have contribution to the enhancer capacity of the CNS2. But why did authors exclude regions IV and V? Actually we can observe a significant increase of activity for complete CNS2 including regions I-V compared to regions I-III only (p < 0.001). 

(2) In Figure 3, I+Oct had very high enhancer activity than most of the other tests, especially than Oct only (p < 0.001). Does it mean that region I is necessary for Oct1 to promote CD69? This sounds conflict with results reflected by Figure 2 that region I didn't increase CD69 promoter transcriptional activity. 

(3) Why did authors not test Oct1+RUNX1 or Oct1+III by using luciferase?

(4) In Figure 3, RLU of the region II only was a little bit higher than those of Oct only or III+MyoD+GATA. The region II only also had a very small deviation (no error bar shown). When compared with Prmo, why was the p-value of II only (p < 0.05) not as significant as the latter, Oct and III+MyoD+GATA, which were less than either 0.01 or 0.001?

(4) Figure 4 shows that silencing RUNX1 could reduce CD69 expression. But this can't prove that the potential BS of RUNX1 on CNS2 did play such a role.

(5) More details are needed for ChIP-seq data presented in Figure 1? Which factor(s) and which cell lines were collected for analysis in this work?

A few minor concerns:

(1) Reference is needed for line 43-45 on page 1 "We previously described …"

(2) GSEA # are needed for ChIP-seq data.

(3) Figure 2B, the first row "VI" should be corrected to "IV".

In general, several important pieces were missed in experiments of Figure 3. The authors rushed to conclusions based on limited evidence/experiment data.  

Reviewer 2 Report

The work of Fontela and collaborators aims at identifying DNA regulatory regions involved in the CD69 transcription regulation. The work is well organized, and experiments are done to respond to specific questions derived from previously described results. I have some questions and suggestions that, according to me, may improve the quality of the work.

Major revisions:

1.    Reference citations in the text should be revised because indicated in different ways (e.g. [9] [10] [11] and not [9-11] or [12] [13] and not [12,13] or [14] [14,16] and not [14-16]. I suggest to use a reference manager software to include references in the text.
2.    Please include in the introduction the importance of CD69 indicating its function also derived from CD69-deficient mice. Moreover, indicate eventually known interactors to better describe the importance of studying how CD69 is regulated.
3.    Please include Sanger sequencing results as supplemental tables to allow to perform a blast/blat analysis of the cloned region and clearly understand which genomic region is considered.
4.    I am a little puzzled regarding the primer sequences. In fact, controlling amplicons produced by in silico PCR implemented in the UCSC Genome Browser and using mm9 as reference genome I obtained an amplicon only associating I+II+III_Fw with I+OCT_Fw primers. Pleas indicate which primers were used together.
5.    Please include in table 1 the expected dimension of the amplicon with primers used together.
6.    How was done siRNA sequence identification? Because the siRNA “Pou2f1.13.1” does not match with Oct1 but with a genomic region coding for CD247 (using mm9). A different result occurs using mm10. This makes more important to indicate which genome release Authors used for their experiments (see minor revisions).
7.    Authors did not check the efficacy of each siRNA on target genes. Please provide the down-regulation of target genes obtained with each siRNA.
8.    In the experiments of siRNA silencing Authors did not confirm the down-regulation of the targeted genes. Please provide this data, for example, using a qRT-PCR or WB approach to associate to figure 4.
9.    What are the shared TFs that bind in the same regions of mouse and human CD69 regulatory elements? Please indicate also this data in the 3.1 result section to improve the knowledge of these regions. Please also indicate in a supplemental table TFs that bind each region. This could be useful for further studies.
10.    Authors say: “Thus, the 3’untranslated region, promoter, CNS1 and CNS2 of both species are enriched in bound transcription factors”. Did they perform a statistical analysis to say that these regions are enriched? If not I think that it is better to say that these regions have the ability to bind many different TFs.
11.    Which statistical test was performed to say “mouse CNS2 also shows the greatest enrichment of TF binding.”?
12.    Authors say: “while those of regions I, IV and V are not essential”. I think that this in an incorrect conclusion because enhancing activity of the region I+II+III+IV+V is the highest. Probably region I is the region lesser important, but authors did not represent data with only that region in figure 2.
13.    In figure 2A, please, indicate which TF bind to conserved sites.
14.    What does it mean the colours in the boxes indicating TFs in figure 2? Please specify this in the figure caption.
15.    How the Authors defined the BSs clustering? (line 168).
16.    Regarding results described in figure 3, please include a comment also regarding the position of each region respect to the promoter. In fact, in the different constructs, the original distance from the promoter of the II and III regions is not maintained.
17.    Figure 3B: include SD in the PMA treatment of the region II.
18.    I do not agree with the following sentence: “The ~40 bp sequence of region II containing the Oct1 TFBS (Prom+Oct) significantly increases the inducible enhancer activity in a similar manner as region II (Prom+II), while the ~60 bp sequence containing GATA+MyoD TFBSs (Prom+MyoD+GATA) did not significantly increase the inducible enhancer activity”. Please rewrite it considering that the higher induction Authors have is when I and Oct1 are in a correct distance from the promoter. Moreover, enhancers III+MyoD+Gata, Oct, and II that do not respect the original organization showed similar results in term of transcriptional enhancing after PMA induction. I think that a more correct experiment is to identify the lesser enhancer dimension with sufficient enhancing activity without mixing DNA organization but producing constructs with smaller and smaller DNA enhancing structures.

Minor revisions:

1.    Please write in vivo in italic font.
2.    Please indicate the version of the genome considered. mm10 or mm9? This information will allow checking genome region using the UCSC genome browser.
3.    CNS2enhancer: add a space between CNS2 and enhancer (line 119).
4.    Please enlarge the dimension of 0 and 1 numbers in figure 1 because at the actual dimension they are not visible.
5.    Authors say: “As shown in Figure 2.B regions II+III have an inducible enhancer activity similar to that of Regions I-III and complete CNS2, while regions I+II have lower enhancer potential.” According to what showed in the figure 2B, PMA induces similarly luciferase activity with I+II+III and II+III CNS2 regions. Different results are obtained with I+II+III+IV+V or I+II. Please reformulate better the phrase to avoid miss-understanding describing regions always in the same way.
6.    Please, correct figure 2B. Authors indicated VI region instead of IV.

Round 2

Reviewer 1 Report

The manuscript has been improved according to the suggestions/comments of the reviewers.

Reviewer 2 Report

Minor comments:

Line 93: "Below is displayed..." instead "Below is display..."

Please, include a statistical significance in Figure 4B.

Author Response

Dear editor,

Please find the point by point response to reviewers below.

Reviewer #2

Minor comments:

1) Reviewer 2 found a grammatical mistake in line 93. We have corrected it accordingly. (page 3 line 93).

2) Reviewer 2 asked for the statistical significance in figure 4.B. We have included it both in the figure and in the figure legend. (page 9-10, figure 4.B and lines 252-253).
